# The HOME Study: Understanding How College Students at a Hispanic Serving Institution Coped with Food Insecurity in a Pandemic

**DOI:** 10.3390/ijerph182111087

**Published:** 2021-10-21

**Authors:** Miriam Manboard, Cassandra M. Johnson, Hannah Thornton, Lesli Biediger-Friedman

**Affiliations:** Nutrition and Foods Program, School of Family and Consumer Sciences, Texas State University, San Marcos, TX 78666, USA; mcm151@txstate.edu (M.M.); c_j216@txstate.edu (C.M.J.); ht1074@txstate.edu (H.T.)

**Keywords:** access to food, food assistance, safety net, nutrition policy, nutrition intervention, university, higher education, college student

## Abstract

College students represent a unique population of adults, who may be more likely to experience food insecurity due to their transient circumstances, limited access to resources, and increased educational expenses. But little is known about how college students and their households mitigate food insecurity, particularly during a crisis like the COVID-19 pandemic. The Household Observations of Meals and Environments (HOME) Study described how college students in the US utilized multilevel resources, including an on-campus food pantry, to maintain food security during the pandemic. A convenience sample of college students (*n* = 18) were recruited from an on-campus food pantry and provided quantitative and qualitative data through online surveys and in-depth Zoom interviews. Survey data were analyzed to describe sociodemographic characteristics. In-depth interviews were recorded, transcribed, coded, and analyzed thematically to identify emergent themes. Social support and the use of an on-campus food pantry were primary factors in maintaining a food security safety net. Students faced barriers when trying to access federal and state food assistance programs and identified multilevel resources, their food security, and the role of social support as facilitators in their perceptions of food insecurity status and experiences. Findings highlight practical implications for research related to on-campus food insecurity interventions and policies to support food security among college students.

## 1. Introduction

The multilevel societal effects of the COVID-19 pandemic have dramatically increased food insecurity for some population groups in the United States (US) [1,2]. Food insecurity is a major public health issue with nearly 14 million US households lacking access to enough food to maintain a consistent, safe, adequate, and healthy diet throughout the year [3]. Between 2011 and 2019, rates of food insecurity in the US declined overall [3]. However, during times of stress and crisis, food insecurity tends to increase, especially for vulnerable populations [3,4,5]. Research indicates that rates of food insecurity among US households may have increased by as much as 32% as a result of the COVID-19 pandemic [1]. Studies of previous disasters demonstrate that, for many, increased food insecurity continues to be an issue even after the major stressor has concluded [4,6].

Since 2015, 20–50% of US college students have reported experiencing food insecurity—A rate that is approximately four times higher than the general population [7,8,9]. In the US, federal food assistance programs attempt to address the food security needs of vulnerable populations. For example, the National School Lunch Program, School Breakfast Program, Child and Adult Care Food Program, and the Special Supplemental Nutrition Assistance Program for Women, Infants, and Children (WIC) serve low-income children, older adults, and pregnant and post-natal women and their children respectively. Despite their high rates of food insecurity, college students have not previously been considered in the creation of US federal food assistance programs, an omission which may contribute to disenfranchisement and an exacerbation of food insecurity [8,10]. 

Administered by the US Department of Agriculture (USDA), the Supplemental Nutrition Assistance Program (SNAP) aims to augment the food budgets of low-income households. Prior to the pandemic, college students in the US were restricted from participating in SNAP if they were attending their institution “more than half-time” [10,11]. In March 2020, the USDA adjusted eligibility requirements for SNAP to increase college student access [5,11]. In January 2021, the new eligibility requirements were extended by Congress through the Consolidated Appropriations Act of 2021 which provided a broader range of income eligibility and work requirements were relaxed to allow individuals suffering from pandemic-related demands to qualify for this assistance program [11,12]. However, these extensions are expected to end by December of 2021 with previous SNAP restrictions for college students returning. 

While temporary relief efforts have been implemented to provide emergency food assistance for vulnerable populations in the US, including college students, the US still lacks a plan for permanent food assistance for college students. Food insecurity negatively impacts college students’ nutritional intake, academic performance, and health outcomes in both the long-term and short-term [13,14]. Studies on college students and their dietary habits during the pandemic have already found that the pandemic has negatively affected the ability of students to acquire food items [15]. Therefore, it is imperative that more is done to address college student food security. 

Previous studies have used quantitative methods to describe the characteristics of food insecurity on US college campuses as well as US college students’ utilization of on-campus food pantries [16,17]. However, little is known about how college students manage food insecurity during a crisis, such as the COVID-19 pandemic. Additionally, to the knowledge of the authors, no studies have employed in-depth qualitative analyses to describe how college students deal with food insecurity. The Household Observations of Meals and Environments (HOME) Study aimed to comprehensively understand the experiences of food insecure college students during the COVID-19 pandemic with qualitative and quantitative data. The study defined the college students’ food security “safety net,” including resources at governmental, community, university, household, and personal levels. The HOME study consisted of four phases that consisted of an initial survey, interview section, a photo-texting section along with a final survey. The current manuscript utilizes the initial survey and interview portions of the study to attempt to answer the following research questions: (1) What resources and support services comprise the food security safety net for food insecure college students; and (2) how did access to resources and support services, including social support, change for food insecure college students during the COVID-19 pandemic.

## 2. Materials and Methods

### 2.1. Context

The Food Security Learning Community (FLSC) at Texas State University (TXST) conducts research on food security on college campuses to design, implement, and evaluate interventions to address food insecurity, based on the nutrition care process and principles of community-engaged research [18,19]. TXST is a large public Hispanic Serving Institution (HSI) in a relatively small city in South-Central Texas, located between two major cities. In the fall of 2020, TXST had 37,849 students [20]. In terms of race and ethnicity, the breakdown of the student population was 43% White, 39% Hispanic, 11% Black, 2.7% Asian, 0.2% American Indian/Alaskan Native, and 0.1% Pacific Islander/Hawaiian Native in 2020 [20]. Most students were female (59.8% female and 40.2% male) [20]. In 2020, more than 27,000 students received financial aid of some kind [20]. The university also reported the capacity to house 6000 students on campus during a semester [20]. Because of the limited housing on campus, and a large student population, most students live off campus. 

This study complements unpublished FSLC research about the design and implementation of the on-campus food pantry, Bobcat Bounty [21]. In March of 2020, Bobcat Bounty was modified from a client choice pantry to a pre-bagged, curbside distribution model to maintain safety practices recommended by public health officials to mitigate the spread of COVID-19. Through the pandemic, Bobcat Bounty continued to provide clients with a variety of food items, including fresh produce, canned goods, grains, and fresh and shelf-stable protein-rich foods. The HOME Study aimed to understand the experiences of food insecure college students who were clients of Bobcat Bounty during this time of crisis and change.

### 2.2. Overview of The Home Observations of Meals and Environment (HOME) Study

The HOME Study consisted of four phases: (1) an initial survey; (2) in-depth interviews; (3) photo and text elicitation; and (4) a final survey. Figure 1 presents the four phases of the HOME study. The current manuscript describes the methodology of the entire study, reports on data from the first two phases (initial survey and interviews), and describes individual factors related to accessing food within this specific environment during the COVID-19 pandemic. A forthcoming manuscript will report on the last two phases (photo and text elicitation and final survey) and focus on the household level factors related to accessing food during the pandemic. 

### 2.3. Theoretical Framework

The Social Ecological Model (SEM) has been used as a basis for the conceptual framework of health promotion programs since its conception in 1988 [22,23], and can be combined with models that focus on individual behavior to gain a more comprehensive look at the influences on a person’s behavior [22,23]. The SEM allows for an understanding of how different factors affect and are affected by behavior on micro, meso, and macro levels [22,23]. The Health Belief Model (HBM) is a widely used theory to understand the health behaviors of individuals [22,24]. There are six constructs in the HBM that act as predictors of an individual’s behaviors: perceived susceptibility; perceived severity; perceived benefits; perceived barriers; cues to action; and self-efficacy [22,24]. Both the SEM and HBM have been used to assess variables related to a community’s or individual’s susceptibility to food insecurity and nutrition-related health outcomes, as well as to assess responses and intervention strategies by public health entities [25,26]. The HBM can be utilized to expand knowledge of the individual level of the SEM, providing more description of the experiences individuals face when navigating food security.

### 2.4. Recruitment and Retention

The intended sample was currently enrolled (part-time or full-time) students at Texas State University, including undergraduate and graduate students in any major, who were living on-campus or off-campus, and who reported experiencing varying severity of food insecurity. The study utilized a convenience sample of participants recruited from clients of the on-campus food pantry, who had previously participated in the food pantry to acquire food items. During the time of recruitment (the summer 2020 semester), the food pantry served an average of 60 clients per week. 

### 2.5. Data Collection Procedures

The Institutional Review Board (IRB) at Texas State University reviewed and granted approval for this study (Project #7319). The HOME Study was conducted in the Fall of 2020, from mid-August and until early November of 2020, which overlapped with the COVID-19 pandemic. During the time of the fall 2020 semester, instructors at the university had shifted to deliver most courses via remote instruction, though there were some in-person courses in the Fall of 2020. There was substantially less on-campus activity that semester. 

Given the COVID-19 pandemic, all data were collected remotely without in-person interaction, including emailing written informed consent forms, utilizing web-based platforms for interviews, and communicating through email. Details on data collection are presented next in order of completion: initial survey, interview, photo-texting, and final survey.

#### 2.5.1. Initial Survey

The initial survey was self-administered through Qualtrics and consisted of 19 questions, including enrollment status, living situation, food security status, utilization of food assistance programs, and general experiences during the COVID-19 pandemic. Responses from the survey were used to determine eligibility for subsequent phases of the study. Inclusion criteria were that participants needed to be college students enrolled in the fall 2020 semester, were between the ages of 18 and 30, and had participated in the pantry at least once. Participants were excluded if they were not students at the university or were not enrolled during the fall 2020 semester.

The survey included the US Household Food Security Survey Module (US FSSM) six-item short form to assess household food security [27]. The USDA categorizes household food security based on affirmative responses to six items and categorizes four levels of food security: high food security, marginal food security, low food security, and very low food security [25]. Low and very low food security are considered food insecure [3,28]. The reference period for food security was altered from “in the last 12 months” to “over the last semester” to better capture food insecurity for college students during the spring semester, or when the pandemic was declared in the US. After completing the survey, research staff reviewed data and scheduled interviews for eligible participants. A total of 18 participants were brought forward for the in-depth interviews. 

#### 2.5.2. In-Depth Interviews

To gain an understanding of college students’ experiences of food insecurity before and during the pandemic, participants completed semi-structured, in-depth interviews. Interviews were conducted via Zoom (an online video-based application) to comply with social distancing and safety protocols. An interview guide (Figure 2) was created by two members of the research team. The guide contained major questions about the household food environment, food security, and access to resources, including social networks and food assistance programs before and during the COVID-19 pandemic. The methodology and creation of the interview questions were guided by Kvale’s and Brinkmann’s InterViews Third Edition [29] and the theoretical constructs of the SEM and HBM. The average interview time was approximately 45 min, with one interview lasting 30 min and another interview lasting 2 h.

#### 2.5.3. Data Analysis and Interpretation

This manuscript analyzed data from the initial survey and interviews of the HOME Study. Research team members de-identified data prior to analysis. Quantitative data from initial surveys were exported into Excel and used to calculate descriptive statistics for personal and household characteristics. Qualitative data from the interviews were analyzed and interpreted.

Research staff recorded the audio-only portions of the interviews and generated transcripts through the auto-transcription feature of Zoom. Transcripts were checked for accuracy against the audio recording. On average, interview transcripts were 12–14 single-spaced pages. Transcripts were de-identified prior to analysis by removing personal information and assigning the participant’s identification number to each transcript. 

The HOME Study used team coding for qualitative data analysis and interpretation, with two coders and a master coder. The research team created a codebook, based on a combination of a priori codes based on the literature and emergent codes based on the data. For example, the codebook included codes for the key constructs of the theoretical model, including self-efficacy, cues to action, barriers to achieving or maintaining food security like economic and physical access to food, including money, transportation, and employment or work, and facilitators like government, community, and on-campus food assistance, and other resources related to food security 

During the analysis, the research team identified emergent themes related to coping strategies and social support. Inductive codes were added to the codebook for the different forms of social support and specific coping strategies. Coping strategies were defined as actions or behaviors related to food acquisition, preparation or cooking, eating, or storage to either avoid or mitigate effects of food insecurity, The code list consisted of three major themes: Social Support, Access to Food, and Coping Strategies. Support was defined as outside groups that supported the participants in having food or other necessities. Access was defined as the ability of the participant to access resources that provided food or other necessities. Codes were also added for consequences or effects of food insecurity like changes in mental health, physical health, or wellbeing; daily routines or lifestyle behaviors, including physical activity and exercise; or social interactions. 

Before the analysis began, all coders received training in qualitative methods and coding, specifically. The coders and senior faculty met regularly to discuss the coding process. Two coders applied all a priori codes to the transcripts. Coders completed coding manually using Microsoft Excel. After reviewing all transcripts, inductive codes were created and applied to transcripts. Responses were analyzed for recurring themes, statements, and constructs. The master coder monitored the coding process and manually reviewed coded interviews. When they noticed segments were coded differently, the coders resolved discrepancies through group discussion. In addition, the master coder calculated interrater reliability with the Holsti Method [30] and checked interrater reliability on a weekly basis. Intercoder reliability was calculated as 78%. Regular meetings to discuss coding contributed to the rigor.

## 3. Results

### 3.1. Study Participants

Eighteen participants completed the initial survey and the interview; fourteen participants completed all four phases (Figure 1). Table 1 shows the characteristics of the 18 participants—currently enrolled college students between the ages of 18 to 30 years old. All lived off-campus and most participants (83%) were female. More than half identified as Black, Latino/a/Hispanic, Asian, Native American, or a combination of races/ethnicities. All participants were categorized as experiencing high/marginal, low, or very low food insecurity over the last semester [3]. All participants reported reliance on Bobcat Bounty (*n* = 18) for food security. Participants identified their social networks (defined as friends and family, *n* = 13) and the university (emergency funding programs available to eligible students, *n* = 11), as important resources. Participants also reported community-based organizations or agencies that were part of their food security safety net, which were not linked to their roles as students, such as local food banks and food pantries (*n* = 7) and churches and religious organizations (*n* = 4). Coping strategies were further reported through open-ended responses by some participants, providing examples of strategies such as donating plasma (*n* = 2), using unspecified university resources (*n* = 3), and receiving food from their apartment complex (*n* = 1). Responses were further elaborated on during the interview phase of the study. Federal food and nutrition assistance programs, such as SNAP and WIC were only reported by three participants total.

### 3.2. Findings

Three major theme categories were identified: Social Support, Access to Food, and Coping Strategies. Within these three themes, participants demonstrated the constructs of the HBM. The themes of social support and access to food related to their perceived facilitators and barriers for food security. The theme of coping strategies was characterized by the constructs of self-efficacy and cues to action. Throughout the themes of social support, access to food, and coping strategies, participants reported that their behaviors were influenced by their beliefs about susceptibility to adverse health outcomes. With respect to how the pandemic has affected their ability to be food secure, multiple participants relayed concerns about their safety while procuring food.

### 3.3. Social Support

Within participant responses about facilitators, the theme of social support, both informational and instrumental, came primarily from the participants’ friends, families, and surrounding community. The subtheme of social support was reported by every participant to be a factor in facilitating and maintaining food security. When questioned about where they found support in maintaining a food security safety net during college and the COVID-19 pandemic, participants overwhelming responded that social support networks were a major factor in maintaining food security. This reported social support network included roommates, friends, classmates, partners, family members, student organizations, and local churches. Participants repeatedly detailed how receiving social support allowed them to maintain a level of food security: 

“Well, my family, it was a huge help because I was able to turn to them when I had to. When they kicked us out, essentially, from the dorms. They were able to help me move the stuff in such short notice and luckily here (at my apartment) we (roommates) take turns, like grocery [shopping]. We don’t have a list or anything just whoever it’s most convenient for will take the opportunity and go grocery shopping or go to (on-campus food pantry)...We’re very compassionate, we help each other out.” 

The sharing of food and other food-related resources, including utensils, recipes, and appliances, was reported as the primary method of support participants received. Social support for maintaining food security was also denoted in the sharing of information about food resources. Some participants reported receiving financial support from their social support networks as well, primarily from family members. Participant responses of how they became aware of an on-campus food pantry centered around their social contacts and networks.

### 3.4. Access to Food

Within participant responses about barriers, the theme of access was split into the subthemes of physical and economic access and systematic access.

#### 3.4.1. Physical and Economic Access

A common theme among participants’ responses to the barriers they faced in achieving food security was a lack of physical access. This was demonstrated by reporting a lack of access to full-scale grocery stores and the Feeding America-affiliated county food bank due to either a lack of transportation, or the changing of store hours due to the COVID-19 pandemic. Participants also reported being unable to use transportation due to COVID-19 pandemic restrictions or safety concerns. In approximately half of interviews participants cited examples of food restrictions based on food availability and preferences that hindered their ability to maintain food security.

In addition, participants reported significant impacts on their financial security due to the COVID-19 pandemic. These impacts included reduced access to usual sources of reliable income, job loss, and reduced hours. These impacts were reported to be experienced by both participants and people they relied on for help with finances.

#### 3.4.2. Systematic Access

During interviews, participants reported a lack of access to systematic assistance. Participants reported being unable to sign up for assistance resources, including SNAP and pandemic relief assistance. When participants reported not being able to access these systematic resources, it was primarily due to not meeting eligibility requirements or not having knowledge on how to access the resources.

When participants did have access to certain systematic resources, such as the on-campus food pantry, there were still reported barriers. Participants reported some issues with using the food pantry as a resource during the COVID-19 pandemic, specifically, food waste or being unable to utilize food items, lack of knowledge on how to use certain foods, and food preferences. These issues were reported as stemming from the change from a client-choice pantry to a pre-bagged, curbside distribution model. Participants did not report these issues as being major barriers to accessing the food pantry. A lack of physical and systematic access was reported by participants to be a compounding barrier as detailed by one participant:

“I tried to figure (WIC) out, and I went because of the booth lady that’s there sometimes when you’re waiting in line. I talked to her about it, and we tried to set it up and I can’t remember what went wrong with it because we did it like a year ago. And then we tried to do it again during pandemic stuff but we only have one car in the house and my roommate works so he uses the car all day, so it was a lot of trying to figure out how to get on the Bobcat Shuttle with the new times or the bus with the new times and try to figure out how to work that which kinda made it more of a hassle than we could deal with.”

Some forms of systematic access reported by participants were identified as facilitators. Systematic access was found on institutional levels. Particularly through the university’s institutional operational structure. Participants listed financial aid in the form of COVID-19 relief funds and mental health counseling provided by the university as the primary means of systematic support. The on-campus food pantry was given as the primary source of systematic food assistance. Participants reported the food pantry as being a necessary resource for maintaining food security for various reasons, including the accessibility of the pantry both in terms of location and signing-up, provision of fresh fruits and vegetables, adequate safety measures in response to COVID-19. Participants reported that using the pantry allowed finances to go to non-food necessities, and that nutrition education information, utensils, and COVID-19 personal protective equipment acquired during their use of the weekly distributions of the on-campus pantry were helpful.

### 3.5. Coping Strategies

The third major theme of coping strategies was found in responses that demonstrated the constructs of self-efficacy and cues to action. Participant responses were coded for whether and how they demonstrated self-efficacy in finding ways to cope with their perceived barriers and stress. Participants routinely reported skills and behaviors developed during the pandemic to maintain food security, reduce susceptibility to contracting COVID-19, and manage stress.

#### Self-Efficacy/Cues to Action

Learned skills and behaviors that predated the participants’ enrollment in college were a recurring subtheme within coping strategies. Participants reported that these skills and behaviors influenced their ability to maintain food security while adapting to the college experience. Parental and familial pre-college support was reported as an influence on participants’ current knowledge, attitudes, and beliefs, as well as the actions participants took to manage food and other resources. Multiple participants recounted learning to cook and prepare food from their parents and spoke about how this previous knowledge contributed to their ability to cook while attending college. Participants also talked about pre-college experiences with food insecurity and observing their family members, primarily parents, handling different levels of food insecurity. Participants reported how previous experiences with decreased food security or use of food assistance programs contributed to their current knowledge, attitudes, and beliefs about maintaining a food security safety net. Conversely, some participants noted that growing up with a higher level of food security resulted in them struggling to cope with reduced food security during college.

“It’s definitely like, I like to plan ahead, write down what meals I would like to make. Make a list and try to get as cheap as I can and stick to a budget, so I don’t have to worry that much about if I have enough money to eat.”

Participants who felt more susceptible to contracting COVID-19 relayed changing behaviors to minimize their perceived susceptibility. Participants repeatedly reported using their perceived need for safety as a cue to change behaviors in accordance with safety guidelines to lower their risk of infection. When questioned on their ability to cope with stress, some participants reported an increased ability to cope with stress as the pandemic progressed. The reasons for improved coping included learning new stress management techniques, having pets, maintaining connections with family or friends, and previous experience dealing with stressful situations.

### 3.6. Food Insecurity Experiences during the COVID-19 Pandemic 

Participants reported changes in their purchasing and eating habits during the COVID-19 pandemic. In some cases, these changes were significant enough to inhibit their ability to access food. In other cases, participants’ actions were not hindered by safety concerns as they perceived their susceptibility to illness as low or non-existent.

“(Obtaining food) was more unsafe here, in (major city). I felt more comfortable (in college town) but (in major city) there are just too many people. And especially when masks weren’t being implemented and especially when I was around old people, I didn’t want to be around them...It was just the amount of people I guess, I didn’t like.”

Many participants reported factors related to perceived stress and feeling vulnerable to increased stress, particularly towards the beginning of the pandemic, with some participants noting feeling a decrease in their ability to cope with stress. Reports of perceived stress were related to changes caused by the COVID-19 pandemic, particularly changes to normal daily activities, including the shutdown of public events and spaces, an inability to be in the company of others, and changes related to their education. Few participants reported feeling increased stress related to their food security specifically, and instead reported stress related to finances, safety concerns, or pandemic-related changes that then influenced their food security. Approximately half of the participants reported using counseling services either within the university system or from outside sources or using medication to support their mental health.

## 4. Discussion

Using surveys and interviews, the HOME Study aimed to examine individual level barriers and facilitators related to maintaining food security during a time of crisis—the COVID-19 pandemic. The first prominent finding highlights the critical role of the on-campus food pantry to the food security of the students who used it. Unlike food and financial resources provided by the government, participants did not report limited access to the on-campus food pantry. Indeed, participants noted that access to the direct food assistance provided by the food pantry enabled them to allocate financial resources to other basic needs, such as rent. This finding is notable as campus food pantries have become a primary intervention in addressing food insecurity among college students [7]. In contrast, findings from the HOME Study demonstrate that college students do not consider federal programs as part of their food security safety net. For example, only three students reported participation in federal nutrition assistance programs. Despite the primacy of food pantries as an intervention for college student food insecurity, there is no established standard for university-level support or implementation.

Another prominent finding was the importance of social support (informational, instrumental, and emotional) offered by friends, family, roommates, and campus and local community groups. Social support and social networks appeared to be a main facilitator of food security among this population. When college students had social support prior to college, during college, and throughout the COVID-19 pandemic they were better able to maintain and navigate a food security safety net. The sharing of food resources and information about food resources was commonly reported by participants as a means of obtaining food. All participants mentioned receiving some form of social support in maintaining their food security, and that social support appeared to be influenced by different factors, including culture and familial status. Remarkably, students who reported food security support networks, also reported an increased ability to focus on academics instead of obtaining food. Food security support may increase access and equity for the successful completion of higher education. The social interaction of the college student warrants additional investigation. In the current study, all participants had roommates. Future research to identify levers for increasing food security would benefit from examining the household food environments of college students as they are different than those of other food insecure individuals.

In addition to social support, the structures of the academic institution (e.g., on- and off-campus support services, transportation, student organizations, financial assistance, and counseling/health services) may facilitate food security. Results reported in this manuscript indicate that emergency funding made possible through a university program was an important contributor to food security. In the initial survey, participants reported relying on their apartment complex (or housing organization) and donating plasma to acquire food or money for food. Given the relationships between food insecurity and housing insecurity, and how overlapping insecurities affect health and wellbeing [7], future studies should investigate the role of capacity building and inter-agency relationships related to food, housing, and health, in order to best support students.

College students as a population are routinely underserved by the current US SNAP program. Though pandemic provisions were implemented to include more college students as eligible SNAP participants [11], these changes are temporary and were not tailored to the rapidly changing and evolving needs of college students. It is necessary for existing government assistance programs at the federal and state levels to address the unique experiences of college students by considering the interaction between food insecurity and the surrounding social and physical food environment. Individual level bias may also influence how researchers, practitioners, and policymakers view college students. In US academic institutions, there is limited representation of individuals from socioeconomically disadvantaged backgrounds [31,32]. Previous research has described a “leaky pipeline” and the adversity that more diverse individuals, such as those from racial/ethnic/cultural minorities or low-income backgrounds, face in higher education [31,32]. More privileged individuals, including researchers, practitioners, and policymakers, may have perspectives about college students that do not reflect the reality of their varied experiences and unintentionally make their experiences of food insecurity less visible. 

The traditional image of a US college student—an individual who begins college immediately after graduating from high school, relies primarily on their parents for financial support while they focus on their college classes, and works less than full-time—is outdated [10]. In the US, college students are a distinct and important population of adults that are increasingly coming from lower income and more diverse backgrounds. By 2018, 71% of US college students were considered non-traditional [10]. Most US federal food and nutrition assistance programs, including SNAP and WIC, are designed with the household as the intended target, though the household may range in size from one person to multi-generational. 

This assumption may be one of the precursors of variance in the literature regarding college student food insecurity contributors and rates. On a large, quantitative scale, we may see cases of overinflation [33] compared to non-college attending peers. However, other more descriptive studies have shown that not all college students fit this mold. Based on research findings from this study, college students enrolled at this Hispanic-serving institution, do not fit this traditional view. This sample of college students are fulfilling adult roles—working for pay, caring for children, caring for parents, and completing academic programs. Further research is needed to expand upon our knowledge of food insecurity and crisis response to support students at minority-serving institutions.

Limitations of the study can be attributed to the challenges of conducting research during a pandemic and the descriptive, convenience sampling strategy. First, due to the COVID-19 pandemic, participants likely experienced changes in their personal or household circumstances between the initial survey and interviews. As a result, survey data may have not represented their circumstances at the time of the interview. Second, survey data were collected through self-report, and participants may have been experiencing increased stress and anxiety associated with the pandemic. This may have impacted participants’ ability to interpret survey items and recall information. Third, research staff recruited participants from clients of the on-campus food pantry. Prior research has documented self-selection into food assistance programs, and described how participants utilizing food pantries, for example, may be different from participants who do not utilize food pantries, even when considering personal or household circumstances [34]. Fourth, as this is a descriptive study, research findings cannot be generalized to all college students without consideration of local contexts and perspectives; however, the sample was consistent with the study’s focus on food insecure college students, and the study provides in-depth information of a unique population during a unique time. 

The use of the US FSSM six-item short form to measure food security is both a limitation and a strength. Ellison and colleagues have presented compelling evidence for improving measurement of food insecurity for college students, specifically [35]. Acknowledging and addressing limitations in food insecurity measurements with college students connects to a larger movement within public health nutrition to expand how food security is measured [34] because there are documented limitations with using a single tool universally, that is, for all applications and populations [36]. For example, items in the FSSM focus more on quantitative deprivation and less on other aspects of food insecurity, like the qualitative (e.g., changes in the food/diet quality), psychological (e.g., feelings of stress and anxiety), and social (e.g., stigma and isolation) domains of food insecurity [37]. Additionally, participants may have not responded affirmatively to the items in the US FSSM [27], because of discomfort or concerns about being judged, which may have underestimated the extent of food insecurity in this sample. Ideally, a scale, developed and validated for college students, would generate the most accurate estimate of food insecurity [35] as Ellison and other scholars have previously described. Use of a supplementary measure of food insecurity like the Four Domain-Food Insecurity Scale (4D-FIS) [37], with this unique subpopulation—college students—may also provide valuable information for developing impactful policy changes. Still, the US FSSM is a valid and reliable tool for surveillance and monitoring at the population-level [38]. In the absence of a population-specific scale for college students, the US FSSM provides a consistent and comparable estimate of food insecurity, which can be compared to other studies [38].

Regarding strengths, the HOME Study, and the research presented in this manuscript, benefited from having qualitative data to help interpret quantitative data. Previous studies on food insecurity among college students have relied primarily on online surveys [16,17]. In the HOME Study, the use of interviews allowed for a more in-depth look at how college students faced food insecurity and maintained a food security safety net during a time of crisis. Qualitative assessments of college student food insecurity and the efforts of college campuses in addressing food insecurity are critical to ensure justice and access to resources for the completion of higher education. 

A second strength was the inclusion of undergraduate and graduate students on the research team. The faculty researchers, student research assistants, and student volunteers on the HOME Study research team worked closely together in design, data collection, and analysis. This contributed to high participant retention through the selection of useful and needed incentives and selection of interactive forms of data collection. The presence of peers (e.g., fellow students) on the research team appeared to increase receptivity and willingness to provide information among participants. Participants seemed more engaged and appreciated being able to share their experiences in a convenient and personal way.

Findings from the HOME Study have implications for practitioners and educators who work with college students, for universities, and for future research on college student food insecurity. College students may not have what they need to achieve or maintain food security and may not feel comfortable asking representatives of the university (e.g., educators or administrators) for support. In this study, college students discussed how essential social support was to maintaining food security; however, study participants primarily identified friends and family as supporters. Educators and other practitioners can help reduce stigma by acknowledging food security as a reality for many college students and pursuing training, as necessary, to better understand this issue. Furthermore, educators and practitioners can take advantage of student social networks to promote strategies like meal planning and food-sharing as well as resources and government assistance programs that mitigate food insecurity.

At the university level, change is needed in policies and practices to increase student access to financial and other resources that support food security. Results from this study indicated that students benefitted from COVID-relief funds and that the on-campus food pantry was an important part of students’ food security safety net. Universities can examine cost of attendance metrics to ensure students have access to appropriate amounts of aid and provide training to help faculty and administrators identify and address food insecurity when it is experienced by students. Insofar as there is a need on campus for emergency food assistance, universities can provide funding and structural support to on-campus food pantries. Finally, by recognizing food security as an essential component of student wellness and success, universities can become advocates for federal food assistance programs that include college students and young adults and other policies that provide lasting solutions to college student food insecurity.

While the last decade has brought increased research on the quantitative aspects of food insecurity among college students, little is known about their experience of food insecurity, in context of their household environments and roommate dynamics. The HOME Study applied qualitative and quantitative methods to comprehensively understand how college students experience and manage food insecurity within their home environments. Forthcoming manuscripts, analyzing data from photo and text elicitation, and the final survey, will examine which characteristics of the household environment are most relevant to the food security of college students and to determine how best to assess those characteristics.

## 5. Conclusions

The present study suggests that the social support of the food insecure college student may reinforce self-efficacy by providing a personal safety net and help college students maintain food security. Results show that having access to an on-campus food pantry also aids students in coping with food insecurity. Further research is needed on how the unique home environments of college students impact how they do or do not maintain food security. The present study demonstrates a need for on-going inter-agency support services that address consistent access to healthy, affordable, and adequate food among college students. Building a multilevel food security safety net that centers the unique, non-traditional characteristics of current US college students and involves resources at the federal, state, regional, and institutional levels is critical for supporting food security among this population.

## Figures and Tables

**Figure 1 ijerph-18-11087-f001:**
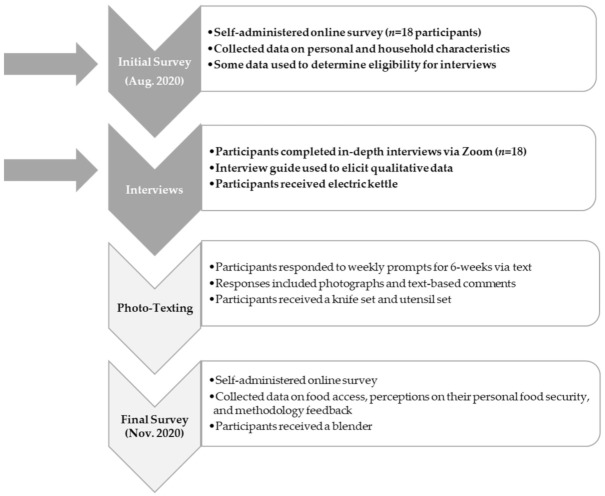
Timeline and research activities of the HOME Study. This figure presents the four phases of the HOME Study. The current manuscript reports on data collected during phase one (initial survey) and phase two (interviews). A forthcoming manuscript will report on the latter phases.

**Figure 2 ijerph-18-11087-f002:**
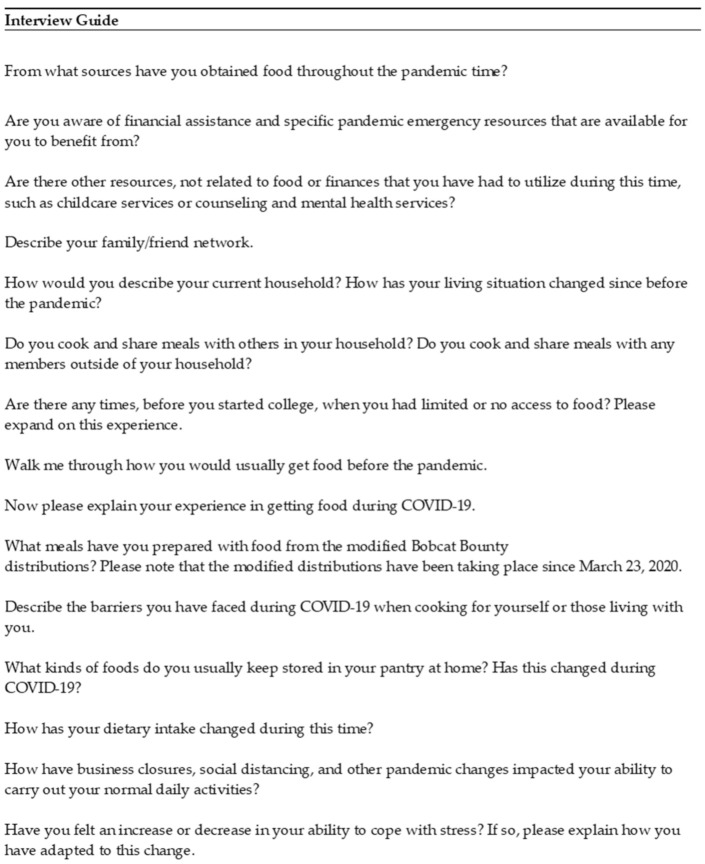
Abbreviated In-Depth Interview.

**Table 1 ijerph-18-11087-t001:** Characteristics of HOME Study participants (*n* = 18).

Characteristic		*n*	%
Age (Years)	18–19	3	17%
	20–23	12	67%
	24–30	3	17%
Gender	Female	15	83%
	Male	3	17%
Race/Ethnicity	Black	2	11%
	Latino or Hispanic	8	44%
	White	5	28%
	Asian	1	6%
	Asian, Latino or Hispanic	1	6%
	Native American, Latino or Hispanic	1	6%
Food Security Status	High/Marginal	3	17%
	Low	13	72%
	Very Low	2	11%
Location	San Marcos	17	94%
	Outside San Marcos	1	6%

This table presents data collected from the initial survey. Personal characteristics, except for food security status, were self-reported. Food security status was determined based on affirmative responses from the U.S. Food Security Survey Module [27].

## Data Availability

The data are not publicly available due to ongoing data analysis for subsequent research manuscripts.

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
