# Peer review of "The HOME Study: Understanding How College Students at a Hispanic Serving Institution Coped with Food Insecurity in a Pandemic"

_ijerph, 2021, doi:10.3390/ijerph182111087_

Round 1
Reviewer 1 Report
Reviewer Comments – Inter J Environ Res Pub Health
- Title could be better …..put the “Hope Study” in the title. It seems hidden until later in the article. Maybe “The HOPE Study: understanding how US college students dealt with food insecurity in a pandemic
- There is a very new publication that needs to be included in the intro and discussion comparisons. “Influence of COVID-19 Pandemic Restrictions on College Students’ Dietary Quality and Experience of the Food Environment.” Large 500 survey using same Six USDA food insecurity survey also using open ended questions that identified changes in shopping and dietary quality measured by HEI. Published in Nutrients August 14, 2021.
- Line 76: This study, the Household Observation of Meals and Environment (HOPE), applied (past tense) mixed methods to understand the experience of food insecure college students during the COVID-10 pandemic. HOME defined (past tense) the college …
- Provide the website link for Bobcat Bounty at Texas State.
- How many students typically access Bobcat Bounty on a regular basis? Did this change with the pandemic? If it increased, how much more were students using the pantry?
- Major: the Photo-texting method is not clear at all. What is this? Students took photos of what? The household food environment – the kitchen, the pantry, ?? How often were they asked…..says over a six week period, but not clear – every week, only when investigator asked, etc?
- Sample size of 18 to start with and 14 to finish is Small. Should be mentioned as a limitation.
- The study was from August to October – so what is the semester being asked about? Summer, Spring Semester? What is “over the last semester” line 157.
- Figure 1 – Interview Guide for what? The Zoom Interviews? Was the list the order and were all of these asked of all participants? If not, this is also a limitation.
- What is “emic data”? line 180. This may be a term used for behavioral studies but not clear to a general reader.
- Line 193 – delete the first sentence and line 196 delete “as described above”.
- Saturation of three is only 21% of the 14 participants.
- How was the intercoder reliability calculated? Did they use a kappa statistic?
- Why didn’t they also have a control group of students NOT using the Bobcat Bounty assistance for comparison?…..another limitation.
- Table 1 consider table title “Characteristics of HOME study participants (n=18)
- This needs to be fixed: the six item food security questionnaire identifies “high or marginal” – not jus “marginal”:
Food security status is assigned as follows:
- Raw score 0-1—High or marginal food security (raw score 1 may be considered marginal food security, but a large proportion of households that would be measured as having marginal food security using the household or adult scale will have raw score zero on the six-item scale)
- Raw score 2-4—Low food security
- Raw score 5-6—Very low food security
- Line 300 – what does “including food waste” refer too? Were they not using the food given – and therefore wasting food? Not clear.”
- Line 324 – not clear…..when were the utensils given to the students?
- If the authors report the % of food insecure at about 80%, what % of the students were “stressed”? Use of counseling services or medication (about 50%) does not give a real measure of stress. This is also in the conclusions…
- Line 397 – “emergent adults” is not clear for more general reader, lingo.
- Line 439 – donating plasma for money? Or donating blood?
- Line 468 – which may have underestimated OR Overestimated. They did not use the two-question screener before using the Six Item which tends to reduce the number of individuals identified as food insecure.
- Limitations – include lack of a comparison control group (non-pantry users)
Author Response
Thank you for your review and comments. Attention was given to all suggestions and comments. The attached table provides a response to each comment. Revisions are highlighted in yellow throughout the revised manuscript. We appreciate your time and dedication to enhancing this work and the field.

Reviewer 2 Report
I review the manuscript entitled "Understanding how US college students dealt with food insecurity in a pandemic: Implications for on-campus food pantries post-COVID".
Abstract is sufficient. The introduction provides though-provoking information on the matter and sets the field of this paper's research. Food insecurity during crisis has not been thoroughly studied yet. On sentence 80 you should change the expression " The current manuscript answers the research questions ..." and you should refer to it as "The current research makes an attempt to give indicative answers on the next 2 questions: .... " (or something similar), as your convenience sample is relatively low.
Materials and methods are described in details. The process was painstakingly designed.
The results and discussion sections are interesting. I suggest that you present strengths and limitations concisely.
Author Response
Thank you for your review and comments. Attention was given to tighten the strengths and limitations section. The attached table provides a response to each comment. Revisions are highlighted in yellow throughout the revised manuscript. We appreciate your time and dedication to enhancing this work and the field.

Reviewer 3 Report
Thank you for the opportunity to review this manuscript. The authors present a qualitative and quantitative analysis of students’ experiences with food security during the pandemic. I applaud the researchers for collecting data during such a challenging time. I have suggested a number of areas in which the manuscript could be further improved. I would be remised to not mention that it is a bit disappointing that the manuscript only provides data from the first two phases of the study and omits findings from the photo-texting and final survey phases. None the less, with revisions I think the results of this study will be a valuable contribution to the existing literature on FI among college students and would be of interest to readers of IJERPH.
Line 21 – suggest removing diverse because without context diversity is quite subjective
Line 37 – Neither reference 1 or 2 provides data for an increased prevalence of FI among college students due to COVID. Please add an additional reference(s).
Line 47 - I don’t believe either of these references (7 or 8) is the most appropriate here. I think it would be more appropriate to provide references to the systematic reviews reporting FI among college students. The methods of Goldrick-Rab (ref 7) have been critiqued as being flawed and reference 8 is just a narrative review. I’ve provided some references below that may be of use to the authors when revising.
- Nikolaus, Cassandra J., et al. "Food insecurity among college students in the United States: A scoping review." Advances in Nutrition2 (2020): 327-348.
- Bruening, Meg, et al. "The struggle is real: a systematic review of food insecurity on postsecondary education campuses." Journal of the Academy of Nutrition and Dietetics11 (2017): 1767-1791.
- Nazmi, Aydin, et al. "A systematic review of food insecurity among US students in higher education." Journal of Hunger & Environmental Nutrition5 (2019): 725-740.
The authors should also consider critiques in the literature that estimates of college FI may be overinflated or inaccurate – especially when compared not just to the general population but to nonstudents of similar ages and even select minority populations (e.g., Native Americans).
- Gundersen, Craig. "Are College Students More Likely to Be Food Insecure than Nonstudents of Similar Ages?." Applied Economic Perspectives and Policy(2020).
Line 48-52 – some of this text seems tangential. It’s no surprise that college students haven’t been thought of as part of the NSLP or WIC. I would personally cut this text and focus more on the next paragraph of SNAP. SNAP is the primary mechanism through which a college student could receive federal food assistance and it would be worthwhile for the authors to expand on the points made in the next paragraph. Reference 9 is a great report on the barriers to SNAP.
Line 53 – A little confusing as written. You mention that they haven’t been considered but then you discuss how they have been considered within SNAP in the next paragraph.
Line 61 – I think it would be good to mention how eligibility was expanded. Readers (especially those of an international audience) would want to know what aspects of eligibility were loosened.
Line 61 - Possibly even reference the following within this paragraph
- Laska, Melissa N., et al. "Food insecurity among college students: An analysis of US state legislation through 2020." Journal of Nutrition Education and Behavior 53.3 (2021): 261-266.
- Laska, Melissa N., et al. "Addressing college food insecurity: An assessment of federal legislation before and during coronavirus disease-2019." Journal of nutrition education and behavior 52.10 (2020): 982-987.
Line 90 – do you have any references to this research?
Line 91 – can you provide a general description of the types of foods that the food pantry usually provides to students? (e.g., does it contain any F/V? refrigerated items? Only shelf-stable products?)
Line 107 – please add how long participation in the study was to complete all 4 phases
Line 110 – Kuddos to the authors for providing details about the use of a theoretical framework.
Line 130 – what does participated in this instance mean? Having received an item any time in 2020. Was the sample restricted to individuals who participated with the food pantry in Jan-March (onset of the pandemic) or Jan-August (the start of the study)?
Line 130 – Is there an estimate of the number of students served by this pantry in a typical semester or academic year that could be provided as the sample size of the potential recruitment pool?
Line 132 – I prefer to say that participants report their food security status. Participants describe their experiences. We as researchers determine if they are food secure/insecure. Also matches phrasing in line 148
Line 136 – It would be helpful to describe in what stage each of the 4 participants were lost to follow-up.
Line 140 – Since universities had very different reopening plans in Fall 2020, the authors should describe the nature of learning at Texas State Univ. at this time. Were students allowed on campus? Were class held in-person, online, hybrid?
Line 177 – This is an interested addition to the paper. I’d like a little more detail. How often were participants given additional prompts (daily? Weekly?) Over the six weeks, how often were participants encouraged to submit a photo? Would the authors be willing to share the prompts used (similar to Figure 1) as a supplemental file?
Line 187 – Was this final survey a duplicate of the initial survey? Were the questions different or was this used to be a pre-post assessment. How many questions were part of this survey?
Line 152 – I agree with the authors that the US FSSM is valid and reliable for various households in the US; however, several researchers have noted that the FSSM may not accurately represent the unique experiences of college students. It’s fine that the researchers used the FSSM (as this is the best we currently have in the field) however, this should be mentioned in the discussion as a limitation. I’ve provided some references below that may be of use to the authors when revising.
- Nikolaus, Cassandra J., Brenna Ellison, and Sharon M. Nickols-Richardson. "Are estimates of food insecurity among college students accurate? Comparison of assessment protocols." PloS one4 (2019): e0215161.
- Nikolaus, Cassandra J., Brenna Ellison, and Sharon M. Nickols-Richardson. "College students’ interpretations of food security questions: results from cognitive interviews." BMC public health1 (2019): 1-16.
- Ellison, Brenna, et al. "Food insecurity among college students: A case for consistent and comparable measurement." Food Policy(2021): 102031.
Line 249 – Section Header “3.1” is used twice (also appears on line 232
Line 396 – I would disagree that SNAP is designed for households. A SNAP participant could be a single person.
Line 398 – Here I feel like Reference 28 is not appropriate for the claim made.
Line 406-409 – The authors mention that they didn’t examine the off-campus food environment but line 79 mentions that the authors considered the “community” aspect of a person’s food safety net.
Line 419, 444 – The authors should avoid classifying findings as “major”.
Line 469 – I don’t feel that Reference 29 is the most appropriate. Please see the suggested references mentioned with line 152.
Line 488 – Since this was not included in the current paper the authors shouldn’t speak to a potential paper.
Line 493-495. I agree the research team is strong by evidence of this well written manuscript; however, this is not needed text within the manuscript.
Line 502 – recommend changing “matter” for another word. (e.g., has implications for ….)
Line 504 – Interest from whom?
Line 506 – I would disagree here that this is the first study to understand the experiences. Also this study was limited (n=18).
Line 510 – Can the authors elaborate here? What aspects would this assessment tool measure within the environment? Who would use this?
Line 512 – I think this is a misleading claim. The sample was drawn from prior participants of a food bank. It’s no surprise that the prevalence of FI was high. However, this may not reflect on the prevalence among the other thousands of non-participating students on your campus.
Line 516 – I disagree with this statement. I feel most educators (and other practitioners, vague) are ill equipped to discuss food insecurity. I feel we shouldn’t ever normalize food insecurity despite it frequent prevalence on a campus. The goal should always be to minimize this experience by connect students with resources.
Line 523 – I encourage the authors to elaborate here. If a policy maker were to read this, what would you want to policies to do exactly? For example, should the policies you are suggesting expanding eligibility access? Provide more money for recipients?
Line 525 – I think this statement dismisses the prior research that has been done that has identified the barriers students experience with on-campus food pantries (things like poor hours, inconvenient location, understaffed, doesn’t have the food they want)
Line 540-542 – I agree but I feel this is the first time the authors ever mention this “transition”.
Line 542 – Categorize students how?
Title: I think the manuscript fails to really provide “implications” suggest revising the title to better reflect the data the manuscript supports.
(Minor) Line 6-13 – Affiliations are not in appropriate journal format.
Author Response
Thank you for your review and thoughtful comments. Our author team carefully considered each comment and we have provided a response to each in the attached table. Each suggested reference was considered and we were able to incorporate many of them. The new figure 1 was added to help clarify the content reported in this manuscript. Substantial revisions were made to the discussion, which we believe enhances the scope of the work and adds context to the current dialog of the field. All revisions to the manuscript are presented in yellow. Your consideration of our work is greatly appreciated.

Reviewer 4 Report
This study uses a convenience sample of 18 students from a college in Texas to examine their food insecurity experiences during COVID-19. The manuscript provides some potentially interesting insights on an understudied topic and population.
I was surprised to read that half of ALL college students have experienced food insecurity? I am not sure that the sources references actually said that. I would actually be very surprised if that was the case.
It is not very clear to me that this study is a mixed methods one. The quantitative analysis is very incomplete and lacks details. I don’t understand what was analyzed, the outcome(s) of interest and whether any statistical analysis was done.
Author Response
Thank you for your review and thoughtful comments. We have replied to each comment in the attached table. All revisions to the manuscript are presented in yellow highlight. Your consideration of our work is greatly appreciated.

Round 2
Reviewer 3 Report
Thank you for the opportunity to review this manuscript again. I think the authors have made thorough edits throughout which have improved the clarity of the manuscript. I have suggested a number of areas in which the manuscript could be further improved. Most of these comments center around (1) I think the author could do a better job of highlighting the unique aspects of their campus environment. That's such a strength of the study and is currently not spotlighted. (2) The discussion needs better organization. There are repeated sections. The general order could be improved as paragraphs jump from topic to topic. The flow is also awkward as authors discuss perspectives of college FI right up front in the discussion rather than discussing their own results. With revisions I think the results of this study remain valuable contribution to the existing literature on FI among college students and would be of great interest to readers of IJERPH.
The title is improved, but I think it could be better. As written “US college students” could make it seem like the sample was nationally representative or that it was conducted on multiple campuses when in fact this was just one southwestern campus and 14 students. I think a strength of the study is that your university is a Hispanic-serving institution. Students at your institution have unique experiences. I would recommend adding that to the title. “The HOME Study: Understanding How College Students At A Primarily Hispanic-Serving Institution Coped With Food Insecurity In A Pandemic”
Section 2.2 – Improvements have been made to make it much more clear what is presented. Greatly appreciated. There is still some confusion based on the terminology the authors use. Line 107 says there are 2 phases. Line 120 says there are 4 phases. I think line 107 should be revised to say there are 4 phases (I wouldn’t group initial survey and interviews together nor photo-texting and final survey. Treat each as its own phase). It should be made clear that the first two study phases (in grey in Fig 1) are discussed in the manuscript and phases 3 and 4 are forthcoming.
Line 152 – This should be mentioned in the discussion (might possibly be a limitation).
Line 232 – Please provide a reference to the Holsti Method
Line 240 – It is interesting that all students live off-campus. I think there needs to be a larger description of the institution within the methods. A potential reader needs to understand basic descriptive of what the campus is like (ethnicity and gender breakdown, urban/rural campus, campus student population, % living on campus) because that greatly influences interpretation of results for generalizability. Suggest adding some of this text to section 2.1.
Discussion: I appreciate all of the additions to the discussion. I think they discuss much of the context and potential limitations (e.g., measurement error). However, the discussion could be better organized. It takes 7 paragraphs before the authors finally discuss the results of the present study (lines 395-449). I would suggest the authors reorganize to state the findings of the study first then discuss then in context of the literature then discuss potential limitations with how we measure FI or describe FI experiences. The first two paragraphs are also somewhat redundant and could probably be condensed. The authors state twice that policymakers, researchers, and practitioners have varying perspectives about college students. Some text throughout the discussion could also be reviewed for duplication. For example line 419-424 are somewhat duplicated with text discussed in line 508-524.
Line 420 is not correct. Reference 31 is written by Ellison and colleagues. Also check line 518.
Line 468 – This is the first mention of students coping by donating plasma or relying on housing organizations. This should be first mentioned within the results section. How many students reported using these methods?
Line 500 – I’m not sure if I’ve ever seen someone say “limited in transferability”. Suggest changing to generalizability.
Line 507 – I like the addition of mentioning your university is a Hispanic-serving institution. Suggest mentioning this additionally earlier in the paper (methods ~Line 92).
Author Response
We appreciate your continued dedication to assist us in improving this manuscript. We have taken the suggestions you have provided and addressed each, making significant modifications to the order and flow of the discussion section. Please see the attached table of our responses. In the revised manuscript, specific changes are highlighted in yellow. However, the discussion section had substantial reorganization and thus we highlighted the section heading.
Thank you.

Reviewer 4 Report
I have no additional comments to make.
Author Response
We appreciate your review and the time you have put into helping us improve this manuscript. We have continued to revise the paper to streamline the discussion section and add clarification in the method section. Thank you for your support.